# Effective Perturbations by Phenobarbital on *I*_Na_, *I*_K(erg)_, *I*_K(M)_ and *I*_K(DR)_ during Pulse Train Stimulation in Neuroblastoma Neuro-2a Cells

**DOI:** 10.3390/biomedicines10081968

**Published:** 2022-08-13

**Authors:** Po-Ming Wu, Pei-Chun Lai, Hsin-Yen Cho, Tzu-Hsien Chuang, Sheng-Nan Wu, Yi-Fang Tu

**Affiliations:** 1Department of Pediatrics, National Cheng Kung University Hospital, College of Medicine, National Cheng Kung University, Tainan 70101, Taiwan; 2Institute of Clinical Medicine, College of Medicine, National Cheng Kung University, Tainan 70101, Taiwan; 3Education Center, National Cheng Kung University Hospital, College of Medicine, National Cheng Kung University, Tainan 70101, Taiwan; 4Department of Physiology, College of Medicine, National Cheng Kung University, Tainan 70101, Taiwan; 5Institute of Basic Medical Sciences, College of Medicine, National Cheng Kung University, Tainan 70101, Taiwan

**Keywords:** phenobarbital (phenobarbitone, Luminal Sodium^®^), voltage-gated Na^+^ current, *erg*-mediated K^+^ current, M-type K^+^ current, delayed-rectifier K^+^ current, pulse train stimulation, γ-aminobutyric acid type A receptors

## Abstract

Phenobarbital (PHB, Luminal Sodium^®^) is a medication of the barbiturate and has long been recognized to be an anticonvulsant and a hypnotic because it can facilitate synaptic inhibition in the central nervous system through acting on the γ-aminobutyric acid (GABA) type A (GABA_A_) receptors. However, to what extent PHB could directly perturb the magnitude and gating of different plasmalemmal ionic currents is not thoroughly explored. In neuroblastoma Neuro-2a cells, we found that PHB effectively suppressed the magnitude of voltage-gated Na^+^ current (*I*_Na_) in a concentration-dependent fashion, with an effective IC_50_ value of 83 µM. The cumulative inhibition of *I*_Na_, evoked by pulse train stimulation, was enhanced by PHB. However, tefluthrin, an activator of *I*_Na_, could attenuate PHB-induced reduction in the decaying time constant of *I*_Na_ inhibition evoked by pulse train stimuli. In addition, the *erg* (*ether-à-go-go*-related gene)-mediated K^+^ current (*I*_K(erg)_) was also blocked by PHB. The PHB-mediated inhibition on *I*_K(erg)_ could not be overcome by flumazenil (GABA antagonist) or chlorotoxin (chloride channel blocker). The PHB reduced the recovery of *I*_K(erg)_ by a two-step voltage protocol with a geometrics-based progression, but it increased the decaying rate of *I*_K(erg)_, evoked by the envelope-of-tail method. About the M-type K^+^ currents (*I*_K(M)_), PHB caused a reduction of its amplitude, which could not be counteracted by flumazenil or chlorotoxin, and PHB could enhance its cumulative inhibition during pulse train stimulation. Moreover, the magnitude of delayed-rectifier K^+^ current (*I*_K(DR)_) was inhibited by PHB, while the cumulative inhibition of *I*_K(DR)_ during 10 s of repetitive stimulation was enhanced. Multiple ionic currents during pulse train stimulation were subject to PHB, and neither GABA antagonist nor chloride channel blocker could counteract these PHB-induced reductions. It suggests that these actions might conceivably participate in different functional activities of excitable cells and be independent of GABA_A_ receptors.

## 1. Introduction

Phenobarbital (PHB, phenobarbitone, phenobarb), also known by the trade name Luminal Sodium^®^, is a medication belonging to a group known as barbiturates. PHB has long been recognized to be an anticonvulsant and a hypnotic because it can facilitate synaptic inhibition in the central nervous system by acting on the γ-aminobutyric acid (GABA) type A (GABA_A_) receptors [1,2,3,4,5,6,7]. GABA_A_ receptor is a ligand-gated chloride ion channel, which is the most common inhibitory channel in the brain. Pentobarbital also belongs to barbiturates and is similar to phenobarbital. It is recently disclosed to suppress neurogenic inflammation and exert neuroprotective and even anti-neoplastic activities by modifying membrane ion channels other than chloride channels [8,9,10]. Membrane ion channels mediate the excitability of membrane potentials, which is essential to the excitable cells, ex, the neurons. Several neurological diseases, including epilepsy, migraine, movement disorders, or muscular disorders, are caused by the dysfunctions of membrane ion channels [1,11]. Thus, we are curious whether PHB might also exert additional actions on other membrane ionic currents in addition to GABA_A_ receptor-mediated chloride currents.

In this study, membrane Na^+^ and K^+^ currents are of interest. As we know, the voltage-gated Na+ (NaV) channels are important for generating and propagating action potentials (APs) in excitable cells [12]. Upon rapid depolarization, Na_V_ channels can go through rapid transitions from the closed (resting) state to the open state and then swiftly change to the inactivated state [12,13]. The inactivation of voltage-gated Na^+^ current (*I*_Na_) has been demonstrated to accumulate before being stimulated during repetitive short depolarizing pulses [14,15]. Several kinds of K^+^ currents were included. First is the *erg* (*ether-à-go-go*-related gene) -mediated K^+^ current (*I*_K(erg)_) gated by voltage-dependent K^+^ (K_V_) channels of EAG (*ether-à-go-go*) family. *I*_K(erg)_ has been known to be intrinsic in cardiac cells or variable types of excitable cells, such as neuroblastoma or neuroendocrine cells. The erg K_V_ channels have peculiar gating kinetics, i.e., rapid inactivation and slow deactivation kinetics. These characteristics are essential in maintaining resting potential and modifying the subthreshold excitability or rhythmic oscillations [16,17,18,19,20,21]. The time-dependent decay of *I*_K(erg)_ evoked by the envelop-of-tail test has also been modified by different small molecules (e.g., azimilide, opioid agonists, and isoplumbagin) [21,22,23,24]. Therefore, it is worthy of investigating to what extent PHB itself can modify the magnitude and gating of *I*_K(erg)_.

The *KCNQ2*, *KCNQ3*, or *KCNQ5* gene is viewed to encode the core subunit of the K_V_7.2, K_V_7.3, or K_V_7.5 channel, respectively. The activity of these K_V_ channels can generate the macroscopic M-type K^+^ current (*I*_K(M)_). The biophysical properties of *I*_K(M)_ are known to exhibit current activation in response to low-threshold voltage and to display a slowly activating and deactivating time course of the current [25,26]. Furthermore, the magnitude of *I*_K(M)_ can potentially regulate the availability of Na_V_ channels during prolonged high-frequency firing [27]. Notable studies have also shown that the presence of either the benzodiazepine activator or cannabidiol could activate K_V_7-encoded currents [28,29]. Therefore, further investigations are warranted to delineate whether PHB has any perturbations on this type of K^+^ currents.

The K_V_ channels play a role in determining the membrane excitability associated with delayed-rectifier K_V_ channels. The activity of K_V_3 (*KCNC*) or K_V_2 (*KCNB*) channels and the magnitude of delayed-rectifier K^+^ current (*I*_K(DR)_) are correlated with AP firing in many cell types [18,26,30]. This type of *I*_K(DR)_ activated during pulse train (PT) stimulations have the propensity to induce the resurgent K^+^ tail current, which is proposed to serve as a negative-feedback mechanism for the closure of K_V_ channels during high-frequency firing [31]. However, the extent to which PHB can modify the magnitude of *I*_K(DR)_, especially during pulse train (PT) stimuli, still remains unclear.

In the current study, the electrophysiological effects of PHB on these membrane ion currents were extensively investigated in Neuro-2a cells. Neuro-2a cells have been previously reported to express functional GABA_A_ receptors with barbiturate binding sites [32]. These membrane ion currents, including *I*_Na_, *I*_K(erg)_, *I*_K(M)_, and *I*_K(DR)_, were explored with a single voltage-clamp pulse or pulse train stimulation and were synergistically inhibited by PHB. This PHB-induced synergistic inhibition of ionic currents may partially contribute to the underlying mechanisms through which PHB or other barbiturates affect the electrical behaviors of excitable cells in cell culture and in vivo.

## 2. Materials and Methods

### 2.1. Chemicals, Drugs, and Solutions

Phenobarbital (PHB, Luminal Sodium^®,^ Solfoton^®^, Tedral^®^, phenobarbitone, phenobarbitol, 5-ethy-5-pheny-1,3-diazinane-2,4,6-trione [IUPAC name], 5-ethyl-5-phenylbarbituric acid, C_12_H_12_N_2_O_3_), flumazenil (FLM), tetraethylammonium chloride (TEA) and tetrodotoxin (TTX) were purchased from Sigma-Aldrich (Genechain, Kaohsiung, Taiwan). Thiopental sodium (Pentothal^®^) was acquired from SCI Pharmatech (Taoyuan, Taiwan), while midazolam was from Nan Kuang Pharmaceutical (Tainan, Taiwan). Chlorotoxin was kindly provided by Professor Dr. Woei-Jer Chuang (Department of Biochemistry, National Cheng Kung University Medical College, Tainan, Taiwan) [21].

For cell preparations, culture media, fetal bovine serum, L-glutamine, and trypsin/EDTA were supplied by HyClone^TM^ (Thermo Fisher, Tainan, Taiwan). All other chemicals were acquired from regular commercial chemicals and of reagent grade. In the current study, reagent water was obtained from a Milli-Q ultrapure water purification system (Millipore; Merck, Tainan, Taiwan).

### 2.2. Cell Preparation

Neuro-2a (N2a), a clonal cell line originally derived from mouse neuroblastoma, was purchased from the Bioresources Collection and Research Center ([BCRC-60026, https://catalog.bcrc.firdi.org.tw/BcrcContent?bid=60026 (accessed on 10 Jan 2022)], Hsinchu, Taiwan). Neuro-2a cells, originally derived from the American Type Culture Collection (ATCC^®^ [CCL-131^TM^]; Manassas, VA, USA), has been used as electrically excitable cells in many studies of electrophysiology and pharmacology [11,33,34,35,36]. Cells were cultured in DMEM supplemented with 10% (*v*/*v*) heat-inactivated fetal bovine serum, 2 mM L-glutamine, 1.5 g/liter sodium bicarbonate, 0.1 mM non-essential amino acids, and 1.0 mM sodium pyruvate in a humidified atmosphere of CO_2_/air (1:19) at 37 °C [21,37,38]. Subcultures were made by trypsinization (0.025% trypsin solution [HyClone^TM^] containing 0.01% sodium, *N*,*N*,-diethyldithiocarbamate and EDTA). Electrophysiological measurements were commonly conducted when cells reached 50–70% confluence (usually 5–7 days) [21].

### 2.3. Electrophysiological Measurements

In the few hours before the experiments, we dispersed Neuro-2a cells with a 1% trypsin/EDTA solution, and a few drops of cell suspension (10^6^/mL) was quickly transferred to a custom-made chamber firmly mounted on the working stage of a DM-IL inverted microscope (Leica; Major Instruments, Kaohsiung, Taiwan). Cells were then bathed at room temperature (20–25 °C) in normal Tyrode’s solution, whose composition is described above. Before each experiment, cells were allowed to settle on the chamber’s bottom. The patch pipettes were pulled from Kimax^®^-51 borosilicate glass tube (#DWK34500–99; Kimble^®^, Merck, Tainan, Taiwan) and were further polished to reach their resistances ranging between 3 and 5 MΩ. During the recordings, the electrodes were mounted in an air-tight holder, which had a suction port on the side, and a silver-chloride wire was used to make contact with the internal electrode solution [39]. We recorded varying types of ionic currents (i.e., *I*_Na_, *I*_K(erg)_, *I*_K(M)_, and *I*_K(DR)_) with the whole-cell mode of a modified patch-clamp technique by using either an Axoclamp-2B (Molecular Devices, Sunnyvale, CA) or an RK-400 amplifier (Bio-Logic, Claix, France), as described elsewhere [19,21,26,40]. The liquid junction potentials that occur when the composition of the pipette internal solution differed from that in the bath were zeroed shortly before giga-Ω formation was made, and the whole-cell data were corrected [21]. As pulse train (PT) stimulation was applied to the tested cell, we used an Astro-Med Grass S88X dual output pulse stimulator (Grass; KYS Technology, Tainan, Taiwan).

### 2.4. Data Recordings

The data were stored online in an ASUSPRO-BU4011LG laptop computer (ASUS, Tainan, Taiwan) at the sampling rate of 10 kHz. The computer was equipped by a Digidata^®^ 1440A acquisition device (Molecular Device; Bestgen Biotech, New Taipei City, Taiwan), through which analog-to-digital and digital-to-analog conversions were controlled by pCLAMP^®^ 10.6 software (Molecular Devices). Current signals were low pass filtered at 3 kHz. We analyzed off-line the signals acquired during the experiments by use of different analytical tools, including LabChart^TM^ 7.0 program (AD Instruments; Gerin, Tainan, Taiwan), OriginPro^®^ 2021 (OriginLab Corp.; Scientific Formosa, Kaohsiung, Taiwan), and custom-made macros built in Excel^®^ 2022 (Redmond, DC, USA). We created various voltage-clamp protocols used in this work from pCLAMP^®^ 10.6 and, through digital-to-analog conversion, different waveforms were used to investigate the current versus voltage (*I-V*) relationship, the steady-state inactivation curve, or the recovery of current inactivation for specific ionic currents (i.e., *I*_Na_ or *I*_K(erg)_) [12,21].

### 2.5. Whole-Cell Data Analyses

In order to evaluate the percentage inhibition of PHB on the *I*_Na_ amplitude, each tested cell was 30 ms depolarized from −100 to −10 mV, and current magnitudes during cell exposure to different PHB concentrations were measured and compared at the start of the voltage pulse. The concentration of PHB required to suppress 50% of the peak component of *I*_Na_ activated in response to short depolarizing pulse was thereafter determined using a Hill function:y=Emax{1+(IC50nH/[PHB]nH)}
where *y* = percentage inhibition (%); [*PHB*] = the PHB concentration applied; *n_H_* = the Hill coefficient; *IC_50_* = the concentration required for a 50% inhibition of *I*_Na_; *E_max_* = PHB-induced maximal inhibition of peak *I*_Na_.

### 2.6. Curve-Fitting Approximations and Statistical Analyses

Curve parameter estimation was made by a non-linear or linear fitting routine with a least-squares minimization procedure, in which the “Solver” add-in bundled with Excel^®^ 2021 (Microsoft, Redmond, WA, USA) was employed [41]. The results are presented as the mean ± standard error of the mean (SEM). The sizes of independent samples (*n*) indicated the cell number from which the investigation was performed. The statistical significance between the two groups was analyzed by using paired or unpaired Student’s *t*-test, while as the difference among more than two groups was needed, post hoc least-significance tests were further performed. The statistical analyses were performed by using SPSS 20.0 software (IBM Corp., Armonk, NY, USA). A statistical significance was considered when *p* < 0.05.

## 3. Results

### 3.1. Effects of PHB on the Amplitude of Voltage-Gated Na^+^ Current (I_Na_) in Neuro-2a Cells

In the beginning, the *I*_Na_ amplitude activated in response to rapid membrane depolarization was tested in different concentrations of PHB. Cells were placed in Ca^2+^-free, Tyrode’s solution, which contained 10 mM tetraethylammonium chloride (TEA) and 0.5 mM CdCl_2_. TEA and CdCl_2_ are recognized to block voltage-gated K^+^ and Ca^2+^ channels, respectively. Upon the short depolarizing pulse from −100 to −10 mV, the peak amplitude of *I*_Na_ with a rapidly activating and inactivating time course was decreased at the exposure to 30 and 100 µM PHB (Figure 1A). For example, in the presence of 100 µM PHB, *I*_Na_ amplitude was profoundly reduced to 812 ± 25 pA (*n* = 8, *p* < 0.05) from a control value of 1867 ± 78 pA (*n* = 8, Figure 1A). The current amplitude was returned to 833 ± 29 pA (*n* = 8) after the removal of PHB. The time constants of *I*_Na_ activation, and fast and slow components of current inactivation in the control period (i.e., no PHB) were respectively 0.61 ± 0.01, 0.91 ± 0.01, and 1.93 ± 0.03 ms (*n* = 8), while those of activation, and fast and slow components of inactivation in the presence of PHB were respectively 0.61 ± 0.01, 0.92 ± 0.01, and 1.94 ± 0.03 ms (*n* = 8, *p* > 0.05). It indicates no obvious change in activation or inactivation time course of *I*_Na_ activated by abrupt membrane depolarization was demonstrated as cells were exposed to PHB.

The relationship between the PHB concentrations and the peak *I*_Na_ was further constructed. As demonstrated in Figure 1B, the *I*_Na_ amplitudes obtained at different PHB concentrations were collated and then estimated. The cumulative addition of PHB in the range between 3 µM and 1 mM resulted in a concentration-dependent reduction in the peak amplitude of *I*_Na_. According to a modified Hill equation stated in Materials and Methods, the IC_50_ value for the PHB-mediated inhibition of peak *I*_Na_ was optimally yielded to be 83 µM. The data, therefore, reflect that the PHB can exert a depressant action on the depolarization-activated *I*_Na_ in a concentration-dependent fashion in Neuro-2a cells.

### 3.2. PHB-Induced Increase in Cumulative Inhibition of I_Na_ Inactivation

High-frequency action potential transmission is important for rapid information processing in the central nervous system. *I*_Na_ inactivation has been previously shown to accumulate prior to being activated during repetitive pulse train (PT) stimulation [14,15,40,42]. Therefore, further experiments were performed to explore whether PHB could modify the inactivation process of *I*_Na_ elicited by the PT stimuli. The PT stimuli were induced by repetitive PT depolarizations to −10 mV (20 ms in each pulse with a rate of 40 Hz for 1 s) when the cell was held at −80 mV. In Figure 2(Aa),B, the *I*_Na_ inactivation was evoked in Neuro-2a cells by a 1 s PT stimulus from −80 to −10 mV with the single-exponential time constant (τ) of 104 ± 28 ms (*n* = 7) in the control period (i.e., no PHB). It indicates a considerable slowing in current inactivation during PT stimulation. When cells exposure to PHB (30 or 100 µM), the exponential time course of *I*_Na_ evoked by the same PT stimulations was strikingly shortened to 81 ± 22 ms (*n* = 8, *p* < 0.05) or 45 ± 14 ms (*n* = 8, *p* < 0.05), respectively (Figure 2(Ab),B), in addition to a profound decrease in *I*_Na_ amplitude in response to rapid membrane depolarization. Furthermore, tefluthrin (Tef, 10 µM) could partially reverse the PHB-mediated reduction of current decay with a τ value of 87 ± 23 ms (*n* = 8, *p* < 0.05, Figure 2B). Tef is a synthetic insecticide known to activate *I*_Na_ [43,44,45].

### 3.3. Effects of PHB on erg-Mediated K^+^ Current (I_K(erg)_) Residing in Neuro-2a Cells

We continued to explore the possible perturbations of PHB on the magnitude of *I*_K(erg)_. In order to amplify *I*_K(erg)_, cells were placed in a high-K^+^, Ca^2+^-free solution containing 1 µM TTX. The recording electrode was filled up with a K^+^-enriched solution [18,19,46]. The different command voltages ranging between −120 and −10 mV for 1 s were imposed to induce hyperpolarization-activated *I*_K(erg)_ when the tested cell was held at −10 mV (Figure 3A) [18,19,21,26]. The average current-versus-voltage (*I-V*) relationship of the peak (filled symbols) or sustained (open symbols) component of *I*_K(erg)_ in the absence (control) or presence of 300 µM PHB was constructed, respectively (Figure 3B,C). For example, upon membrane hyperpolarization from −10 mV, PHB decreased the peak *I*_K(erg)_ from a control value (i.e., no PHB) of 686 ± 156 pA (*n* = 8) to 343 ± 112 pA (*n* = 8, *p* < 0.05) at the level of −60 mV and decreased the sustained *I*_K(erg)_ from 526 ± 82 pA (*n* = 8) to 222 ± 88 pA (*n* = 8, *p* < 0.05). The presence of PHB (300 µM) could suppress both the peak and sustained component *of* deactivating *I*_K(erg)_ throughout the entire measurement (Figure 3C).

We further investigated whether PHB-induced inhibition of *I*_K(erg)_ was associated with GABA-mediated inhibition or not. FLM is an antagonist of GABA_A_ receptors, while chlorotoxin is a blocker of Cl- channels. In the presence of PHB (300 µM), neither FLM nor chlorotoxin could overcome the PHB-mediated inhibition of *I*_K(erg)_ activated by long-lasting membrane hyperpolarization (Figure 3D). The results suggested that PHB-induced inhibition of *I*_K(erg)_ in these cells is unlinked to its propensity to interact with GABA_A_ receptors, as reported previously in other cell types [7,9,33,47,48,49,50].

### 3.4. Modification by PHB on the Magnitude of I_K(erg)_ Activated by PT Stimulation

We further explored if PHB could modify the extent of *I*_K(erg)_ activated in the PT hyperpolarizing stimuli. The stimulus protocol, consisting of repetitive hyperpolarization to −110 mV (40 ms in each pulse with a rate of 20 Hz for 1 s), was imposed over the tested cells held at −10 mV. In the absence of PHB (control), the 1 s repetitive hyperpolarization from −10 to −110 mV evoked the *I*_K(erg)_ decay with a decaying τ of 147 ± 22 ms (*n* = 7) (Figure 4(Aa)). Upon exposure to 300 µM PHB, the τ value in the exponential time course of decaying *I*_K(erg)_ evoked by the same train of hyperpolarizing pulses was shorted to 87 ± 11 ms (*n* = 7, *p* < 0.05) (Figure 4(Ab)), and the *I*_K(erg)_ amplitude was reduced, too (Figure 4B). E-4031 is an inhibitor of *I*_K(erg)_ [18,51], and E-4031 could further diminish the *I*_K(erg)_ amplitude in the presence of 300 µM PHB (Figure 4B). These results indicate that, apart from the reduction in *I*_K(erg)_ magnitude, the decrease in the decaying of *I*_K(erg)_ elicited by a 1 s PT hyperpolarizing pulse (i.e., accumulative inactivation of the current) can be enhanced by PHB in these cells.

### 3.5. Slowing in Recovery from I_K(erg)_ Block Caused by PHB in Neuro-2a Cells

We continued to explore if PHB could lead to any modifications on recovery from the block of *I*_K(erg)_. The recovery from the current block was conducted by using a two-step voltage-clamp protocol in which the interval of depolarizing command pulses (i.e., conditioning pulse) varies with a geometrics-based progression. After each conditioning pulse, a hyperpolarizing pulse stepped back to −110 mV for 500 ms was applied to evoke deactivating *I*_K(erg)_. As the duration of the conditioning pulse was set at 2048 ms, the amplitude of the tail current (i.e., deactivating *I*_K(erg)_) activated by hyperpolarizing step from −10 to −110 mV with a duration of 500 ms was taken as 1.0. The relative amplitude of *I*_K(erg)_ with different duration of conditioning pulse was measured and then compared. In the absence of PHB (control), the peak amplitude of deactivating *I*_K(erg)_ was fully restored from the block when the pulse duration was set at 2048 ms or above (Figure 5(Aa),B). Interestingly, the time course of recovery from the current block was appropriately fitted by a single exponential, and the τ values could be derived. The τ values were 387 ± 12 ms (*n* = 7) in the absence of PHB and 823 ± 23 ms (*n* = 7) in the presence of PHB. As a result, PHB produces a considerable lengthening in the recovery from the block of deactivating *I*_K(erg)_ in Neuro-2a cells.

#### Modification by PHB of the Time Course of *I*_K(erg)_ Evoked by the Envelope-of-Tail Test

Earlier investigations have demonstrated a time-dependent change in *I*_K(erg)_ activation during the envelope-of-tail test [21,22,23,24,52]. Here, we then analyzed an inward activating and deactivating *I*_K(erg)_ evoked by varying durations of conditioning pulse (from 4 to 2048 ms with a geometrics-based progression) to −10 mV from −110 mV, and the relationship of the relative amplitude versus the pulse duration was established in Figure 5C. Like earlier investigations [21,24], the envelope-of-tail test tailored for the activation of *I*_K(erg)_ seen in Neuro-2a cells exhibits a time-dependent exponential decay in the ratio of the relative amplitude (i.e., *I*_act_/*I*_deact_) evoked during the pulse durations between 4 and 2048 ms in a geometrics-based progression. When Neuro-2a-cell exposure to 300 µM PHB, the τ value of *I*_K(erg)_ activated by the envelope-of-tail method was significantly reduced to 64 ± 3 ms (*n* = 7, *p* < 0.05) from a control value of 112 ± 11 ms (*n* = 7). It is conceivable that PHB has the propensity to reduce the time course of *I*_K(erg)_ activation evoked during the envelope-of-tail voltage protocol residing in Neuro-2a cells.

### 3.6. Modification by PHB of M-Type K^+^ Current (I_K(M)_) in Neuro-2a Cells

Next, we continued examining if PHB could modify the magnitude of *I*_K(M)_ in Neuro-2a cells. Cells were placed in a high-K^+^, Ca^2+^-free solution, and the recording electrode was filled up with K^+^-enriched solution. Upon membrane depolarization from −50 to −10 mV, PHB-treated cells exerted an inhibitory effect on *I*_K(M)_ amplitude (Figure 6A,B). This PHB-mediated inhibition of *I*_K(M)_ amplitude could not be reversed by adding FLM (10 µM) or chlorotoxin (1 µM) (Figure 6B). Thus, unlike the stimulatory action of benzodiazepine derivative or cannabidiol on K_V_7-encoded currents, PHB exerts a depressant action of *I*_K(M)_ in Neuro-2a cells. However, the inactivating properties of *I*_K(M)_ shown in Figure 6A could be due to the interference by other types of delayed-rectifier K+ currents. The averaged I-V relationships of *I*_K(M)_ with or without PHB (300 μM) were made, and the results are shown in Figure 6C.

### 3.7. Modification by PHB on I_K(M)_ Inactivation during PT Stimulation

A recent study has demonstrated the capability of *I*_K(M)_ to maintain the availability of Na_V_ channels during prolonged high-frequency firing [27]. Therefore, we further examined if the PHB could modify the extent of *I*_K(M)_ during PT stimulation in Neuro-2a cells. The results showed that PHB (100 µM or 300 µM) obviously decreased the amplitudes of activating and deactivating *I*_K(M)_ during 1 s PT stimulation (Figure 7A–C). For example, PHB (300 µM) decreased the amplitudes of activating *I*_K(M)_ to 165 ± 21 pA (*n* = 7, *p* < 0.05, Figure 7B) and the amplitudes of activating *I*_K(M)_ to 459 ± 195 pA (*n* = 7, *p* < 0.05, Figure 7C). These disclosed that PHB-mediated inhibition of *I*_K(M)_ remained efficacious upon high PT stimulation. It is conceivable that the availability of Na_V_ channels during high-frequency firing in unclamped cells became further retarded during the exposure to PHB, despite its ability to suppress the *I*_Na_ amplitude directly, as described above.

### 3.8. Inhibitory Effect of PHB on Delayed-Rectifier K^+^ Current (I_K(DR)_) Residing in Neuro-2a Cells

The *I*_K(DR)_ has been previously revealed to be suppressed by pentobarbital, another barbiturate, as well as to display the resurgent K^+^ tail currents during high PT stimulations [50,53] We additionally explored if the PHB could modify the magnitude of *I*_K(DR)_ in these cells. In this series of experiments, cells were placed in Ca^2+^-free, Tyrode’s solution, which contained 1 µM TTX and 0.5 mM CdCl_2_, and the recording electrode used was filled up with K^+^-enriched solution. Under the voltage-clamp current recordings, the tested cell was held at −50 mV, and a series of voltage pulses ranging between −60 and +50 mV was imposed over it. In Figure 8A, upon the presence of 300 µM PHB, the *I*_K(DR)_ amplitude was clearly suppressed, especially at the voltages above −10 mV. The average *I-V* relationship of *I*_K(DR)_ is illustrated in Figure 8B.

### 3.9. PHB-Induced Increase in Cumulative Inhibition of I_K(DR)_ Inactivation in Neuro-2a Cells

In the last series of measurements, we wanted to investigate if PHB can modify the time course of *I*_K(DR)_ in response to long-lasting PT stimulation intrinsically in Neuro-2a cells. Under the control condition (i.e., no PHB), a single 10 s depolarizing step from −50 to +50 mV resulted in an exponential decline with a time constant of 2.12 ± 0.08 s (*n* = 8) (Figure 9A). In contrast, the time constant for 10 s PT stimulation to +50 mV, each of which lasted 100 ms with 100 ms interval at −50 mV between the depolarizing stimuli, was conceivably reduced to 1.78 ± 0.07 s (*n* = 8, *p* < 0.05, Figure 9(Ba)). In the presence of 300 µM PHB, the decaying time constant in response to 10 s repetitive depolarizing pulses was obviously decreased to 0.92 ± 0.05 s (*n* = 8, *p* < 0.05, Figure 9(Bb)). This decrease in the decaying time constant mediated by PHB could not be modified by FLM (10 µM). Thus, it is possible that the excessive accumulative inactivation of *I*_K(DR)_ during the exposure to PHB might be irrelevant to GABA_A_ receptors.

## 4. Discussion

The noticeable findings demonstrated in this work are that PHB can regulate the magnitude of multiple types of ionic currents (i.e., *I*_Na_, *I*_K(erg)_, *I*_K(M)_, and *I*_K(DR)_) residing in Neuro-2a cells. Consistent with previous observations [54], the presence of PHB itself suppressed peak *I*_Na_ in a concentration-dependent manner with an estimated IC_50_ of 83 µM (Figure 1). However, the magnitude of PHB-mediated block in these ionic currents failed to be overcome by the subsequent addition of FLM or chlorotoxin. The cumulative inhibition of ionic currents activated during pulse train stimulation was facilitated by PHB. The PHB-induced rise in *I*_Na_ decay during repetitive stimulation was reversed by Tef. These experimental observations reflected that the reduction of *I*_Na_, *I*_K(erg)_, *I*_K(M),_ and *I*_K(DR)_ caused by PHB in Neuro-2a cells tends to be acute and may be independent of GABA_A_ receptor-mediated chloride currents. In addition, these actions could potentially contribute to PHB-induced perturbations on the electrical behaviors of excitable cells (e.g., the discharge patterns in the high-frequency firing of action potentials [APs]), presuming that similar in vivo findings appear [55,56].

Previous work has demonstrated that the train of depolarizing pulses could be efficacious in perturbing the magnitude of *I*_Na_, i.e., current decaying over time in an exponential fashion [14]. This blocking effect is essential in regulating sensory transduction because the driven spike rates can reach hundreds of hertz in response to strong stimuli. As a corollary, the reduced availability of Na_V_ channels during prolonged repetitive activity (i.e., long-lasting trains of high-frequency APs) would attenuate the spike waveform, and Ca^2+^ flux and Ca^2+^-dependent exocytosis would be subsequently impaired at the nerve terminal. In our study, PHB was noted to shorten the τ value of *I*_Na_ decay responding to PT stimulation (Figure 2). Thus, it would further diminish Na_V_-channel availability at the same time.

The magnitude of *I*_Na_ could be rapidly declined in an exponential manner during high-frequency stimulation, as demonstrated previously [14,40,42] (Figure 2). In other words, during prolonged repetitive activity, the availability of Na_V_ channels was profoundly decreased in a time-dependent fashion. By comparison, the decaying rate of *I*_Na_ during such high PT stimulation was noticeably faster than that of *I*_K(erg)_, *I*_K(M)_, or *I*_K(DR)_ (Figure 4, Figure 7, and Figure 9) seen in Neuro-2a cells. Although the gating kinetics or inactivation-dependent mechanisms residing in these K^+^ currents are overly distinguishable [21,24,31,46,52], the difference in current decaying during PT stimulation could be of particular significance. The main reason for this is because the magnitude of these K^+^ currents (i.e., resurgent K^+^ tail currents) could maintain the membrane in a spike-ready state, thus conferring optimal repriming of *I*_Na_ during the occurrence of prolonged repetitive activity [27,31]. In this scenario, the augmentation of cumulative *I*_K(erg)_, *I*_K(M)_, or *I*_K(DR)_ inactivation caused by PHB during high-frequency activity is relevant because it is able to suppress the rapid firing of neuronal APs and cause an additional loss-of-change in stable waveform and high-fidelity synaptic signaling, particularly in high-frequency firing [27]. Therefore, during high-frequency firing of neurons, the presence of PHB itself would decrease the magnitude of post-spike and steady-state currents leading to diminishing the subthreshold depolarized potential.

In the current study, PHB could slow the recovery of the *I*_K(erg)_ block and increase the decay of *I*_K(erg)_ activated by the envelope-of-tail method (Figure 5). These findings suggested that the molecule of PHB is capable of interacting with the open states (conformations) of the K_erg_ channels present in Neuro-2a cells.

Distinguishable from a recent study [28], the *I*_K(M)_ present in Neuro-2a cells was directly suppressed by PHB. However, it should be emphasized that the *I*_K(M)_ amplitude can gradually and progressively arise during repetitive APs because of its slow activation and deactivation kinetics [27]. Upon high-frequency stimulation, the accumulation of *I*_K(M)_ is allowed to hyperpolarize the after-potential and consequently speed up the recovery of Na_V_ channels from inactivation as well to increase the availability of these channels. The PHB facilitated the cumulative inhibition of *I*_K(M)_ during PT stimulation (Figure 7). It means that the waveform of neuronal action potentials (APs) in high frequency could be distorted by PHB, and presynaptic signaling occurring across the full dynamic range of the system was thereafter potentially perturbed.

Despite its rapid deactivation kinetics, *I*_K(DR)_ amplitude (e.g., K_V_3.1-encoded current), identified previously as a timely resurgent K^+^ current, can be progressively raised during high-frequency stimulation [31]. During high-frequency stimulation, the accumulation of *I*_K(DR)_ can be allowed to hyperpolarize the after-potential as well as speed up the recovery of Na_V_ channels from inactivation. In this study, we found that the *I*_K(DR)_ in Neuro-2a cells can be susceptible to being blocked by PHB, and the cumulative inhibition of *I*_K(DR)_ during prolonged (i.e., 10 s) repetitive stimulation was enhanced (Figure 9). Consequently, the magnitude of *I*_Na_ recovery from current inactivation during PT stimulation would become slowed owing to the enhanced cumulative inhibition of *I*_K(erg)_, *I*_K(DR)_, and *I*_K(M)_ [27,31,57]. In this scenario, as cells were exposed to PHB, the high-frequency firing of neuronal APs would lose the emergence of APs’ stable waveforms leading to serious perturbations in high-fidelity synaptic signaling (e.g., the presynaptic release of neurotransmitters) [56]. In other words, the negative feedback mechanism on K_V_ channel closure during high PT stimulation would be impaired during the exposure to PHB [27,31]. Therefore, susceptibility to PHB in vivo may depend on the pre-existing resting potential level, the high-frequency firing, and the concentration of PHB.

From previous pharmacokinetic studies, the recommended plasma concentrations of PHB for anti-seizure activities ranged between 10 and 35 µg/mL (or 43 and 151 µM), while for prophylaxis against febrile convulsions was around 15 µg/mM (or 65 µM) [3,5,58]. Because PHB is a lipophilic compound, its membrane concentration would be higher than blood levels. This study showed that the PHB concentration for the half-maximal inhibition of *I*_Na_ seen in Neuro-2a cells was 83 µM, a value within the clinically applied doses. The observed effects of PHB presented herein can achieve clinical concentration, especially for the treatment of seizure activities lined to high-frequency AP firing.

By synergistic inhibition of multiple ionic currents, including *I*_Na_, *I*_K(erg)_, *I*_K(M)_, or *I*_K(DR)_ in a frequency-dependent manner, PHB might induce other unpredictable electrophysiological responses, in addition to solely being an activator of GABA_A_ receptors. The clinical use of PHB should take these effects into account. However, the findings in the current study are limited to Neuro-2a cells. Further studies will have to validate these in other excitable cells or in vivo.

## Figures and Tables

**Figure 1 biomedicines-10-01968-f001:**
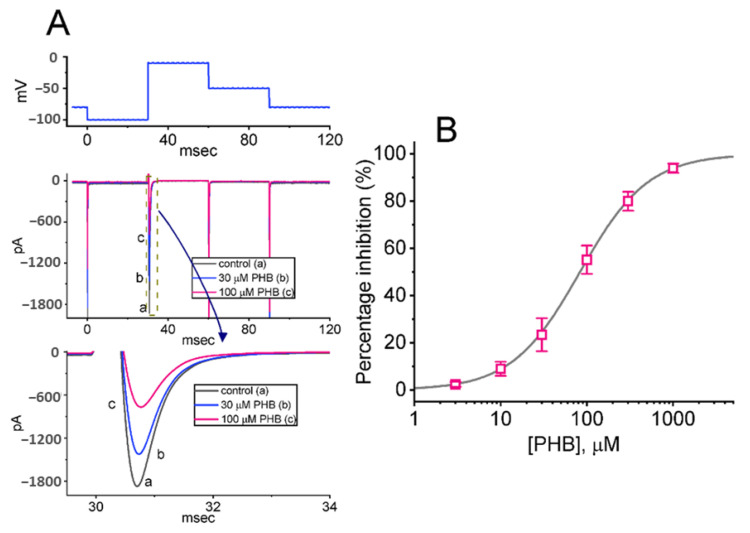
Effects of phenobarbital (PHB) on voltage-gated Na^+^ current (*I*_Na_) in Neuro-2a cells. (**A**) Representative current traces were obtained in the control period (a, absence of PHB, black line) and during cell exposure to 30 µM PHB (b, blue line) or 100 µM PHB (c, pink line). The uppermost part shows the voltage-clamp protocol applied, while the lower part indicates an expanded record from the dashed box in the middle part of (**A**). (**B**) Concentration-dependent response of PHB-mediated inhibition of peak *I*_Na_ residing in Neuro-2a cells (mean ± SEM; *n* = 8 for each point). The sigmoidal gray line, on which the data points were overlaid, indicates the best fit to a modified Hill equation, as mentioned in Materials and Methods.

**Figure 2 biomedicines-10-01968-f002:**
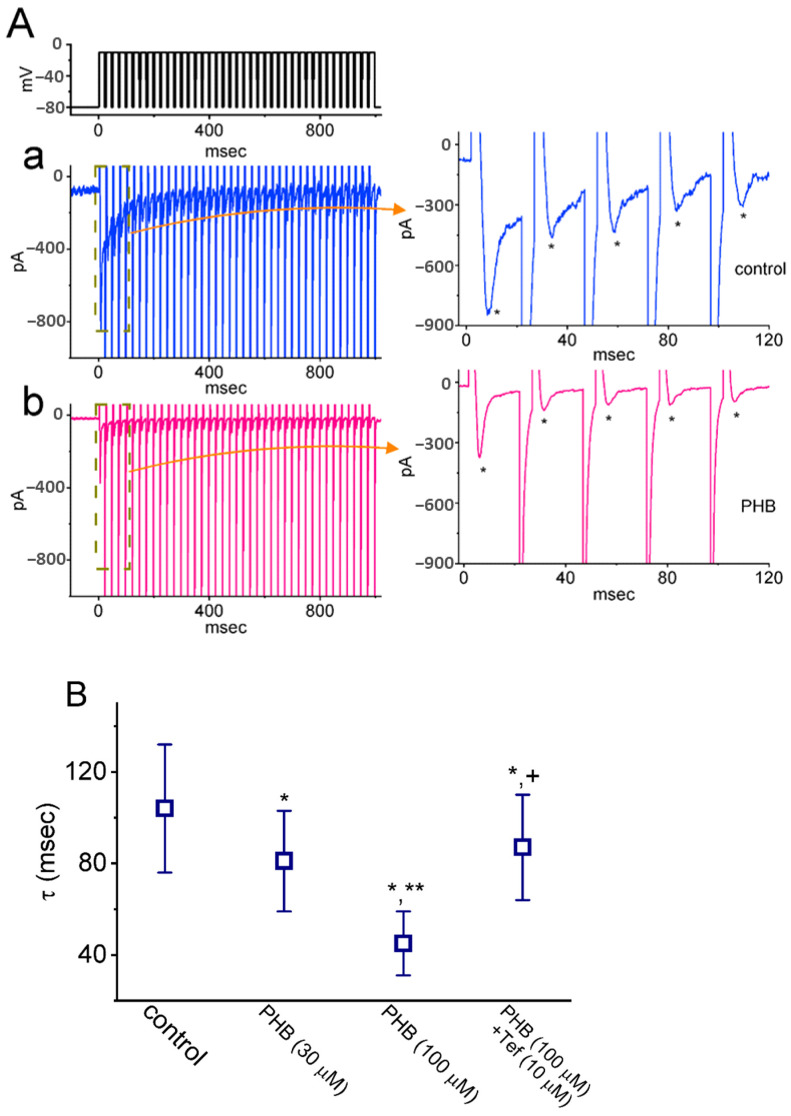
Effect of PHB on *I*_Na_ decline evoked during a 40 Hz train of depolarizing pulses in Neuro-2a cells. The pulse train (PT) stimuli were designed to comprise 40 20-ms pulses which are separated by 5 ms intervals at −80 mV with a total duration of 1 s. (**A**) Representative current traces are taken in the control period (**a**, blue color) and during the exposure to 100 µM PHB (**b**, red color). The uppermost part is the voltage-clamp protocol delivered to the tested cell (i.e., PT stimulation). To provide a single current trace, the right side of (**A**) shows the expanded records from the broken boxes in (**Aa**) and (**Ab**). The asterisks indicate the occurrence of peak *I*_Na_ (i.e., transient inward deflection) activated during PT stimulation. (**B**) Summary graph demonstrating the effect of PHB (30 or 100 µM) and PHB plus tefluthrin (Tef, 10 µM) on the time constant (τ) of *I*_Na_ decay in response to depolarizing PT stimuli from −80 to −10 mV (mean ± SEM; *n* = 8 for each point). Of note, the PHB addition results in a decrease in the τ value of *I*_Na_ activated by PT depolarizing pulses; and the subsequent application of 10 µM Tef can overcome the PHB-induced decrease in τ value. * Significantly different from control (*p* < 0.05), ** significantly different from PHB (30 µM) alone group (*p* < 0.05), and ^+^ significantly different from PHB (100 µM) alone group (*p* < 0.05).

**Figure 3 biomedicines-10-01968-f003:**
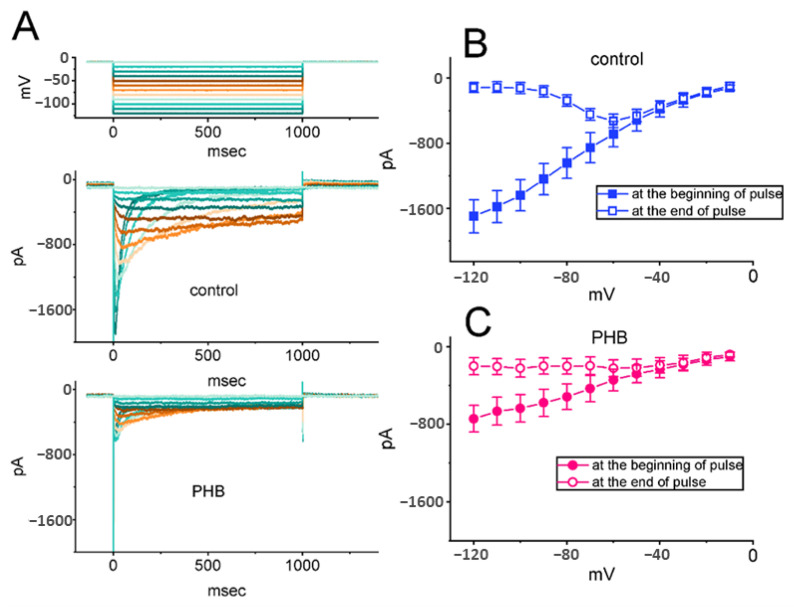
The inhibition of PHB on steady-state *I-V* relationship of hyperpolarization-activated *I*_K(erg)_ and comparisons among effects of PHB, PHB plus flumazenil (FLM), or PHB plus chlorotoxin on *I*_K(erg)_ amplitude identified in Neuro-2a cells. (**A**) Superimposed current traces acquired in the control period (i.e., PHB was not present) (upper) and during cell exposure to 300 µM PHB (lower). The voltage-clamp protocol used is illustrated in the top part, and potential traces shown in different colors correspond with current traces which were evoked by the same level of step commands. In (**B**,**C**), the averaged *I-V* relationships of the peak (filled symbols) or sustained component (open symbols) of *I*_K(erg)_ acquired in the absence (upper, blue color) and presence (lower, pink color) of 300 µM PHB were demonstrated, respectively. Each point in (**B**,**C**) represents the mean ± SEM (*n* = 8). (**D**) Comparison among effects of PHB, PHB plus FLM, or PHB plus chlorotoxin on *I*_K(erg)_ amplitude recorded from Neuro-2a cells (mean ± SEM; *n* = 8 for each point). The current amplitude (i.e., absolute value) taken during exposure to different tested compounds was measured at the start of the 1 s hyperpolarizing step from −10 to −110 mV. * *p* < 0.05 compared with control; ** *p* < 0.05 compared with 100 µM PHB. Of note, neither FLM nor chlorotoxin can reverse PHB-induced inhibition of deactivating *I*_K(erg)_ in Neuro-2a cells.

**Figure 4 biomedicines-10-01968-f004:**
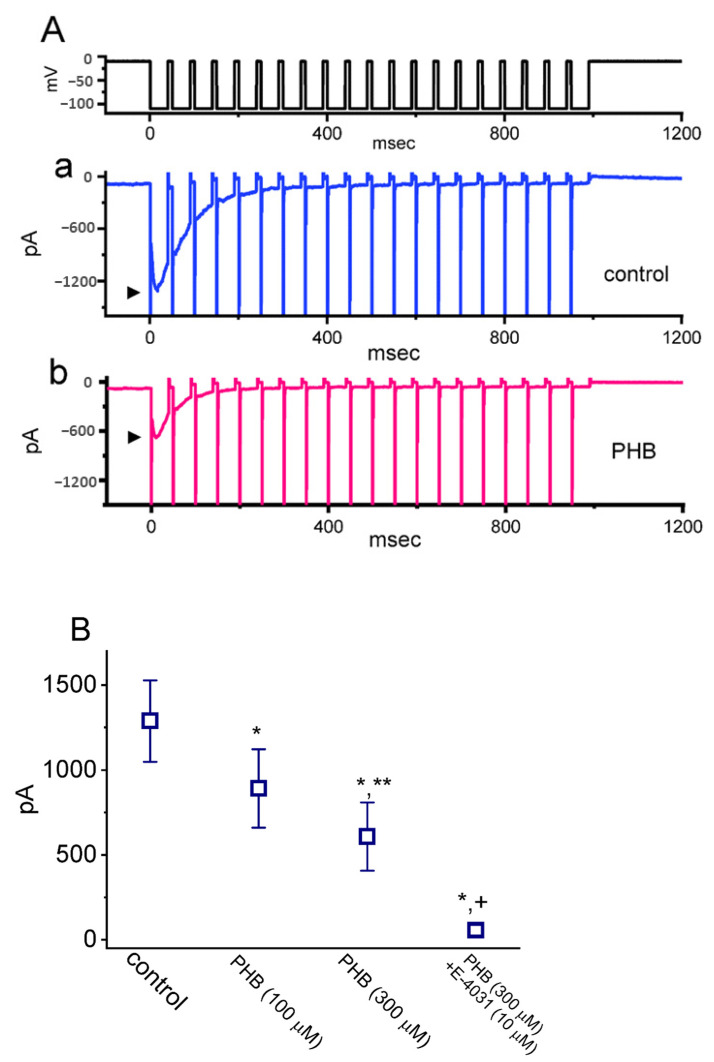
Effect of PHB on cumulative inhibition of *I*_K(erg)_ in response to pulse train (PT) hyperpolarizing stimuli recorded from Neuro-2a cells. In these experiments, we applied a train of hyperpolarizing pulses which consist of 20 40 ms pulse (stepped to −110 mV) separated by 10 ms at −10 mV for a total duration of 1 s. (**A**) Representative current traces taken in the control period (**a**, absence of PHB, blue color) and during cell exposure to 300 µM PHB (**b**, pink color). The top part indicates the voltage-clamp protocol applied (black color), and the arrowhead in each trace shows the peak amplitude of deactivating *I*_K(erg)_ with a resurgent (i.e., hook-of-tail) property. (**B**) Summary graph showing effect of PHB (100 and 300 µM), PHB (300 µM) plus E-4031 (10 µM) on deactivating *I*_K(erg)_ during PT stimulation (mean ± SEM; *n* = 7 for each point). Current amplitude of *I*_K(erg)_ was measured at the start of 1 s PT stimuli from −10 to −110 mV. * Significantly different from control (*p* < 0.05), ** significantly different from PHB (100 µM) alone group (*p* < 0.05), and ^+^ significantly different from PHB (300 µM) alone group.

**Figure 5 biomedicines-10-01968-f005:**
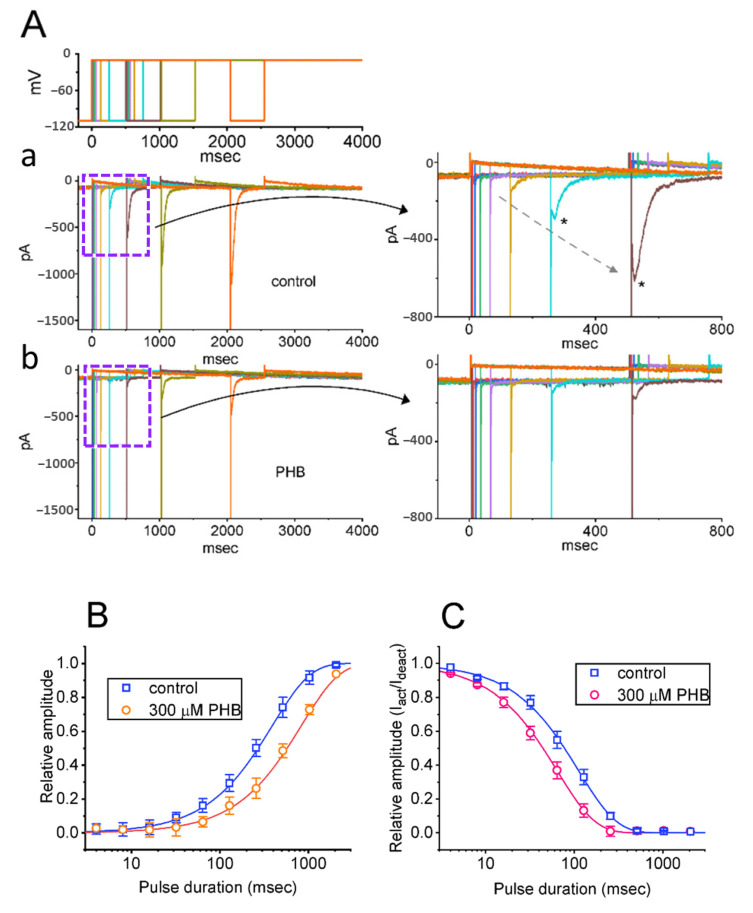
Effects of PHB on both the recovery of *I*_K(erg)_ decay and *I*_K(erg)_ magnitude evoked by the envelop-of-tail test seen in Neuro-2 cells. (**A**) Representative current traces are taken in the control period (**a**. absence of PHB) and during cell exposure to 300 µM PHB (**b**). Each color represents the specific current trace evoked by different voltage pulse with increasing interval of interpulse. The panels shown on the right side indicate the expanded records from purple dashed boxes on the left side for better illustrations. The asterisks show the occurrence of deactivating *I*_K(erg)_, while the gray dashed arrow indicates a gradual increase in peak component of deactivating *I*_K(erg)_ with the prolonged duration of conditioning pulse from −10 to −110 mV. (**B**) Relationship of the duration of conditioning pulse versus the relative amplitude of deactivating *I*_K(erg)_ (mean ± SEM; *n* = 7 for each point). The amplitude of deactivating current at the conditioning pulse with a duration of 2048 ms was taken as 1.0. The blue square symbols are control data points, and the orange circle symbols were acquired during cell exposure to 300 µM PHB. (**C**) PHB-mediated changes in *I*_K(erg)_ magnitude were evoked by the envelop-of-tail test (mean ± SEM; *n* = 7 for each point). The relationship of the pulse duration versus the relative amplitude of *I*_K(erg)_ is illustrated. The relative amplitude appearing at the *y* axis was measured, when the inwardly gradually increasing current (i.e., activating *I*_K(erg)_, *I*_act_) activated by conditioning pulse was divided by peak deactivating *I*_K(erg)_ (*I*_deact_) obtained following a return of the voltage to −110 mV. The decaying rate of *I*_K(erg)_ evoked during the envelop-of-tail occurred in a single exponential function. Of notice, the relationships in (**B**,**C**) are illustrated with a semi-logarithmic plot. The smooth curves with or without the application of 300 µM PHB indicate the best fits to a single exponential function.

**Figure 6 biomedicines-10-01968-f006:**
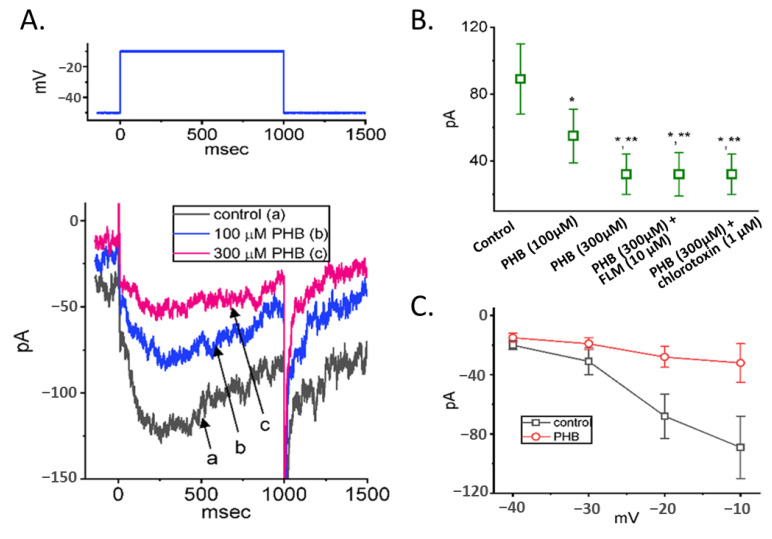
Effect of PHB on M-type K^+^ current (*I*_K(M)_) identified in Neuro-2a cells. (**A**) Representative current traces (i.e., *I*_K(M)_’s) acquired in the control period (a, black line) and during the exposure to 100 µM PHB (b, blue line) or 300 µM PHB (c, pink line). The voltage-clamp protocol we used is in the upper part. (**B**) Summary graph demonstrating effect of PHB (100 or 300 µM), PHB (300 µM) plus flumazenil (FLM, 10 µM), or PHB (300 µM) plus chlorotoxin (1 µM) on *I*_K(M)_ amplitude (mean ± SEM; *n* = 8 for each point). Current amplitude was measured at the end of 1 s depolarizing pulse from −50 to −10 mV. * Significantly different from control (*p* < 0.05) and ** significantly different from PHB (100 µM) alone group (*p* < 0.05). (**C**) Averaged *I-V* relationship of *I*_K(M)_ obtained in the absence (black open squares) and presence (red open circles) of 300 μM PHB (mean ± SEM; *n* = 8 for each point). Current amplitude was measured at the end of each voltage step.

**Figure 7 biomedicines-10-01968-f007:**
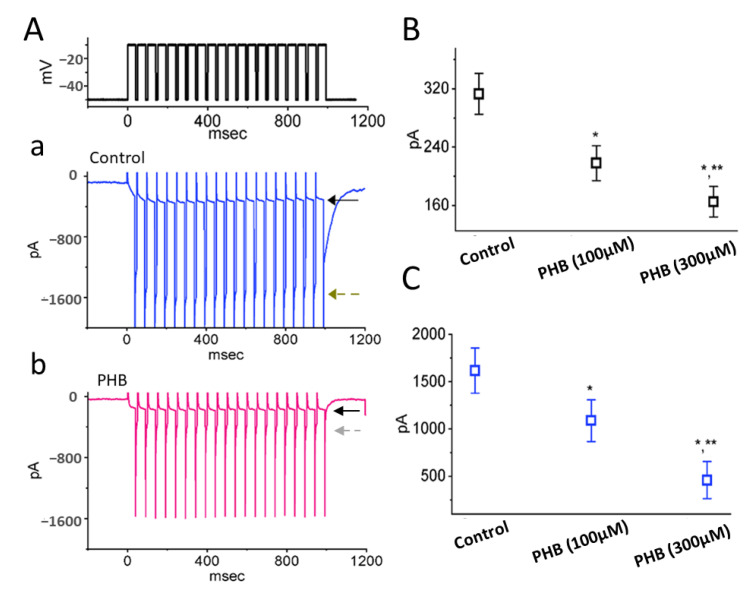
Effect of PHB on *I*_K(M)_ activated by a pulse-train (PT) stimulations identified in Neuro-2a cells. The train was designed to consist of 40 20 ms pulses (stepped to −10 mV) separated by 5 ms intervals at −50 mV for a total duration of 1 s. (**A**) Representative current traces taken during the control period (**a**, blue color) and during the exposure to 300 µM PHB (**b**, red color). The uppermost part shows the voltage-clamp protocol applied. Black solid arrow indicates the activating *I*_K(M)_, while brown dashed arrow is the deactivating component of *I*_K(M)_ obtained following return to −50 mV. Summary graphs in (**B**,**C**), respectively, demonstrate the activating and deactivating amplitudes of *I*_K(M)_ in the absence and presence of 100 or 300 µM PHB (mean ± SEM, *n* = 7 for each point). Activating or deactivating (or tail) amplitude of *I*_K(M)_ was measured at the end of each depolarizing pulse from −50 to −10 mV or following return to −50 mV, respectively. * Significantly different from control (*p* < 0.05) and ** significantly different from PHB (100 µM) alone group (*p* < 0.05).

**Figure 8 biomedicines-10-01968-f008:**
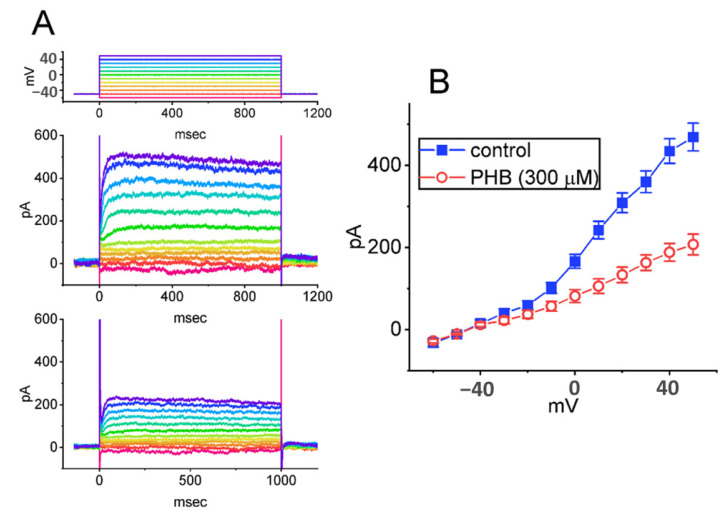
Inhibitory effect of PHB on delayed-rectifier K^+^ currents (*I*_K(DR)_) observed in Neuro-2a cells. (**A**) Superimposed current traces obtained in the absence (upper) and presence (lower) 300 µM PHB. The voltage-clamp protocol applied is illustrated on the top part. Each color represents corresponding current trace evoked by specific voltage pulse. (**B**) Averaged *I-V* relationship of *I*_K(DR)_ taken with or without the application of 300 µM PLB (mean ± SEM; *n* = 8 for each point). Of note, current amplitudes measured at the voltages above −10 mV were suppressed by adding PHB (300 µM). Current amplitude was measured at the end of each voltage step. ▇: control; ○: during exposure to 300 µM PLB.

**Figure 9 biomedicines-10-01968-f009:**
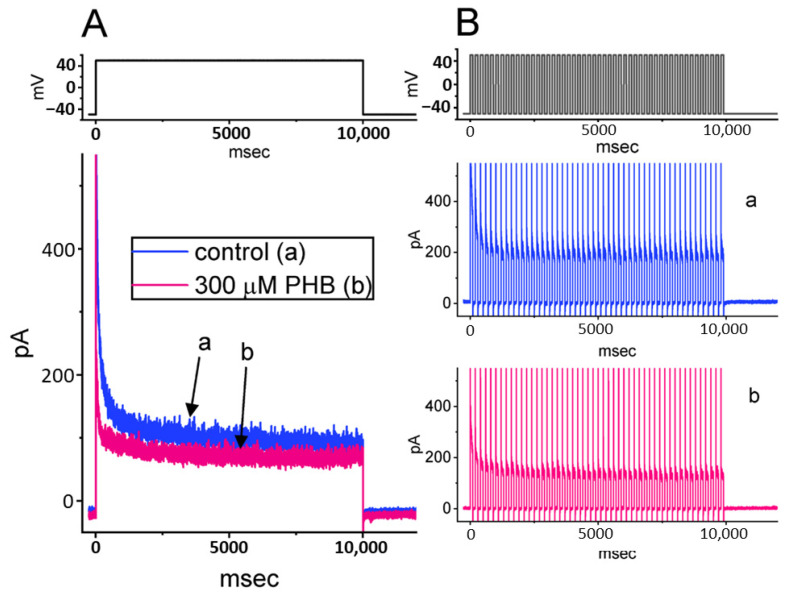
Effect of PHB on long-lasting *I*_K(DR)_ taken with or without the application of PHB (300 µM) inherently residing in Neuro-2a cells. (**A**) Representative current traces (i.e., *I*_K(DR)_’s) were obtained in the absence (a, blue color) and presence (b, pink color) of 300 µM PHB. The tested cell was depolarized from −50 to +50 mV with a duration of 10 s, as indicated in the upper part of (**A**). (**B**) Excessive accumulative inactivation of *I*_K(DR)_ during repetitive stimuli in the absence (upper, **a**, blue color) and presence (lower, **b**, pink color) of 300 µM PHB measured from Neuro-2a cells. Ionic currents were acquired during 100 ms repetitive depolarizations from −50 to +50 mV with a total duration of 10 s (indicated in the upper part of (**B**)). Of note, in addition to the inhibition of *I*_K(DR)_ amplitude, the presence of PHB increases the rate of excessive accumulative inactivation of *I*_K(DR)_ activated by repetitive stimuli.

## Data Availability

The original data are available upon reasonable request to the corresponding author.

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
