# Peer review of "Effective Perturbations by Phenobarbital on INa, IK(erg), IK(M) and IK(DR) during Pulse Train Stimulation in Neuroblastoma Neuro-2a Cells"

_biomedicines, 2022, doi:10.3390/biomedicines10081968_

Round 1
Reviewer 1 Report
This manuscript reported on the Phenobarbital (PHB) can modify the magnitude of membrane ionic currents, especially during PT stimuli, which is an interesting topic for various neurological or psychiatric disorders. However, it is my opinion that this manuscript in its present form is not yet ready for publication. In favor of improving the paper, some points are listed below:
1. Without Plagiarism software, I can clearly see that many texts are just copied and pasted from the internet, which is not good for the journal.
2. The authors should revise the English as well as sentence construction of the manuscript thoroughly. Some texts are difficult to understand. For example,
a. Line number 73, “Owing to its unique biophysical properties (i.e., rapid inactivation and slow deactivation kinetics)...” What does it mean?
b. Line number 24, “…in the continued presence of PHB, further addition of tefluthrin could reverse PHB-induced reduction in the decaying time constant of INa inhibition during pulse train stimuli…”
3. It is nice to explain why it is important to understand the role of the magnitude of membrane ionic currents in neurological disorders (refers to the other published articles).
4. Abbreviations should be defined at first mention in each of the following sections in your paper. It is not the case in the section Abstract for ‘erg-mediated K+’
5. Authors concluded that ‘..the reduction by this compound of INa, IK(erg), IK(M), and IK(DR) in Neuro-2a cells tends to be acute and is independent of its binding to benzodiazepine (GABAA) receptors...’’ It is difficult to conclude without looking for the expression of GABAA, either by western blotting or immunohistochemistry.
6. What is the exact cell density used for the electrophysiological measurements? It is not mentioned in the manuscript.
7. Is this study only specific to Neuro-2a cells or any kind of neuroblastoma because it is difficult to judge only with one cell type?
Reviewer 2 Report
In this manuscript, the authors studied the effects of phenobarbital (PHB) on different ion channels in neuroblastoma Neuro-2a cells during pulse train stimulation. These ion channels include voltage-gated Na channels and voltage-gated K channels. They found that PHB inhibited the Ina, Ikr, Ikm, and other Kv currents. During pulse train stimulation, the inhibition of each ion current is enhanced. The data seem convincing, and the conclusions seem appropriate. But there are a few concerns that need to be addressed below.
Major concerns:
1. Line 206, where is the data showing the current amplitude after the removal of PHB? Please add the trace to Figure 1A if possible.
2. Line 207, is there any data to support no change in the activation and deactivation kinetics?
3. Line 246, what’s the potential mechanism behind the rescue of PHB-induced current decay by tefluthrin? In other words, do they share the same binding sites to modulate the Ina channels?
4. Line 275 and Figure 3 title, the term “hyperpolarization-activated Ik(erg)” is not accurate since ERG channels are actually activated by depolarization rather than hyperpolarization. Inward currents in Figure 3A are the recovery of ERG currents upon hyperpolarization.
5. E-4031 is an ERG channel blocker and blocks the depolarization-activated ERG current, however, here in the study, E-4031 was used to block the recovery of the inactivation of ERG channels upon hyperpolarization. Please explain the reason why E-4031 was used? Have the authors tried recording the IKerg with typical ERG depolarization protocol (doi: 10.1126/science.aaf8070)?
6. It would be better to show I-V relationships for Ikm before and after the application of PHB. Also, in Figure 6A, a depolarization step of -10 is not positive enough to achieve the maximal conductance of M channels. Have the authors tried a more positive voltage?
7. Figure 6A, at -50 mV, shouldn’t three current traces start from the same baseline?
8. Figure 6B, why does there seem to be the inactivation of the M channel at the end of the depolarization phase since M channels are non-inactivating Kv channels?
9. Line 552, Ik(m) do not inactivate.
Minor concerns:
1. In Figure 3D, the Y-axle should be negative currents.
2. Grammar errors: Line 326, 532, and 538.
3. 10 mM FLM and 1 mM Chlorotoxin showed no effects on the inhibition by PHB addition, could it be because the concentration was too low?
4. Please choose better colors for figures, some are confusing—for example, Figures 3A, 5A, and 8A.
Round 2
Reviewer 2 Report
No more comments.
Author Response
Reviewer 2.
Q. No more comments.
A: Thank you very much for the opportunity to revise our manuscript.